# Coping Strategies Furthering Post-Traumatic Growth in Multiple Sclerosis: A Longitudinal Study

**DOI:** 10.3390/ijerph191912679

**Published:** 2022-10-04

**Authors:** Irene Gil-González, Agustín Martín-Rodríguez, Rupert Conrad, María Ángeles Pérez-San-Gregorio

**Affiliations:** 1Department of Personality, Assessment, and Psychological Treatment, University of Seville, 41018 Seville, Spain; 2Department of Psychosomatic Medicine and Psychotherapy, University Hospital Muenster, 48149 Muenster, Germany

**Keywords:** multiple sclerosis, post-traumatic growth, coping strategies

## Abstract

(1) Background: Patients’ behavioral attempts in dealing with Multiple sclerosis (MS) play an important role in post-traumatic growth (PTG). In a longitudinal study, we aimed to identify coping strategies predicting PTG. (2) Methods: 260 MS patients answered the Post-traumatic Growth Inventory and the Brief COPE Questionnaire at three time points during a 36-month follow-up period. (3) Results: an interaction effect between PTG level and assessment time was found for emotional support, positive reframing, active coping, and planning coping strategies. Positive reframing, emotional support, instrumental support, religion, planning, and self-distraction positively predicted PTG. (4) Conclusions: to encourage PTG development, early interventions in MS patients are recommended to promote adaptive coping, particularly positive reframing, social support, active coping, planning, religion, and self-distraction.

## 1. Introduction

Multiple sclerosis (MS) is a chronic neurodegenerative disease impacting a wide spectrum of patients’ life domains, as well as physical, psychological, and social well-being [1,2,3]. Unpredictable progression and a highly heterogenous symptomatology pose a great challenge for affected patients, which also may lead to positive psychological changes [4]. Inner growth after struggling with a life-threatening situation is denominated “post-traumatic growth” (PTG). The term “post-traumatic” means that growth is the consequence of a critical life event. “Growth” refers to a mental and emotional inner state above the initial level [5].

Factors underlying PTG have been widely explored in the scientific literature, with special emphasis on coping strategies. Conceptually, coping strategies are defined as cognitive and behavioral efforts to manage specific external and/or internal demands [6].

In general, problem-focused coping as well as avoidance coping styles, have been related to PTG, [7,8] highlighting the complexity of coping with situational demands to constructively process a trauma. In people living with serious medical conditions, sharing negative emotions and cognitive processing are associated with positive health outcomes. Patients and survivors reporting higher PTG use more frequently adaptive coping strategies, such as problem-focused coping, positive reappraisal, seeking emotional support, and acceptance [9]. Moreover, it has been suggested that problem-focused and social support strategies play an important role in promoting personal growth among people with physical disability [10]. The following coping strategies were identified as contributing factors of PTG in specific health conditions: problem-focused active coping, which involves direct actions to deal with the stressor, in cervical dystonia [11]; acceptance and putting into perspective in hemodialysis [12]; emotional engaging in hematopoietic cell transplantation [13]; instrumental and emotional social support in liver transplant recipients [14] and hematopoietic cell transplant recipients [13]. In people with acquired brain damage, self-distraction and humor were associated with higher PTG in addition to active coping, emotional and instrumental support, positive reframing, planning, acceptance, and use of religion/spirituality [15].

With regard to MS there is growing evidence for the relevance of coping and PTG for well-being and mental health. Thus, in a recent study using semi-structured interviews, re-evaluation of life was described as a beneficial consequence of MS experience [16]. In quantitative studies, reappraisal and positive reframing were found to predict PTG [17] and personal benefit finding [18]. However, there is a lack of studies exploring these associations over a longer period. Longitudinal data indicate an increasing progression of PTG in MS and corroborate the influence of clinical, demographic, and mental health variables with small to medium effect sizes [19]. When studying personal benefit finding, an improvement tendency over time was also observed suggesting that substantial improvement of personal growth appears later in MS adaptation [17]. An ideographic study investigating people with MS disease experience highlighted patients’ expression of gratitude for family support enhancement, religious faith promotion, and making a new identity meaning in the long run [20].

Against this backdrop, the present study aimed to investigate the association between post-traumatic growth and coping strategies over the course of 36 months with special emphasis on strategies positively predicting PTG.

## 2. Materials and Methods

### 2.1. Participants and Procedure

Outpatients of Virgen Macarena University Hospital were asked to participate in three different periods: June 2017–May 2018 (T1), December 2018–December 2019 (T2), and May 2020–April 2021 (T3).

Eligibility criteria included: (a) confirmed MS diagnosis according to McDonald criteria verified by their primary care neurologist; (b) aged over 18; and (c) mental, physical, and cognitive capability to fulfill the assessment protocol. Figure 1 represents the sample selection process at the three different periods in different colors.

The research was authorized by the Ethics Committee responsible (0846-N-18). All participants were provided with oral and written instructions and gave their informed consent to take part in the study.

### 2.2. Instruments

Participants completed a standardized form on sociodemographic features. We collected relevant clinical information from a medical data base.

#### 2.2.1. Post-Traumatic Growth

The Spanish version of the Post-traumatic Growth Inventory (PGI-21) assesses patients’ perception of personal benefit after their experience with MS [21,22]. PGI-21 consists of 21 items, scored on a Likert-type scale from 0 (“no change”) to 5 (“very great degree of change”). Items group into the following five dimensions: relating to others, new possibilities, personal strength, spiritual change, and appreciation of life. In the present sample, Cronbach’s alpha was 0.92, 0.91, and 0.93 for the total score scale and 0.74–0.80, 0.77–0.83, and 0.72–0.88 for the five subscale dimensions, at T1, T2, and T3, respectively.

#### 2.2.2. Coping Strategies

The Brief COPE Questionnaire (COPE-28) evaluates different actions in dealing with stressful situations [23]. The COPE-28 contains 28 items scored on a Likert scale from 0 (“I have not been doing this at all”) to 3 (“I have been doing this a lot”). Test results provide information about 14 subscales: (1) acceptance; (2) emotional support; (3) humor; (4) positive reframing; (5) religion; (6) active coping; (7) instrumental support; (8) planning; (9) behavioral disengagement; (10) denial; (11) self-distraction; (12) self-blaming; (13) substance use; and (14) venting. The Spanish version was applied to our sample [24]. Cronbach’s alpha ranged from 0.61 to 0.84 at T1; 0.70 to 0.96 at T2; and 0.63 to 0.95 at T3 for the above-mentioned subscales.

#### 2.2.3. Expanded Disability Status Scale (EDSS)

The Expanded Disability Status Scale (EDSS) is the predominantly used scale to measure disability in MS. Its efficacy and reliability have been proven at every disease stage. EDSS scores range from 0, indicating regular neurological functioning, to 10. The scales assess different functional systems (FS): visual functions, brainstem functions, pyramidal functions, cerefellar functions, sensory functions, bowel and bladder functions, cerebral functions, and ambulation. A 5-point EDSS score means no ambulatory problem. In EDSS scores over 5 points, the ambulation status is the main factor determining the degree of disability [25,26,27].

### 2.3. Statistical Analysis

Descriptive analyses were calculated to report sample characteristics. A one-way analysis of variance (ANOVA) was used for quantitative variables (age, EDSS, months since diagnosis, and months since the outbreak) and a Chi-squared test was used for qualitative variables (gender, partnership, occupation, educational level, and MS subtype) to detect differences in sociodemographic and clinical variables between three subgroups defined by PTG level (low, medium, and high).

A 3 × 3 mixed factorial ANOVA with Bonferroni post-hoc tests was calculated to study the influence of initial post-traumatic growth level (low: ≥36, medium: 37–60, and high: 61–97) on the use of coping strategies in follow-up assessments (T1, T2, and T3).

To identify potential biopsychosocial predictors of post-traumatic growth level, three different stepwise multivariate regression models were calculated. Total post-traumatic growth level at T1, T2, and T3 were considered dependent variables. Coping strategies at T1, T2, and T3 were introduced as predictors.

Statistics were conducted using SPSS Statistics version 26 (IBM, Armonk, NY, USA). Significance level was set to *p* < 0.05. Effect size coefficients were calculated using G*Power Software 3.1 (University of Duesseldorf, Duesseldorf, Germany) and interpreted according to Cohen´s (1988) recommendations for w (0.10 = small, 0.30 = medium, and 0.50 = large), for f (0.10 = small, 0.25 = medium, and 0.40 = large), f2 (0.02 = small, 0.15 = medium, and 0.35 = large effects) and d (0.20 = small, 0.50 = medium, and 0.80 = large effects) [28].

## 3. Results

The final sample was composed of 260 adults with MS. The sample comprised 179 (68.83%) women and 81 (31.27%) men. The mean age was 45 (SD = 10.60). The mean EDSS at T1 was 3.20 (SD = 1.93), and the most reported MS type was relapsing remittent. Participants’ demographics and clinical characteristics are reported in Table 1.

### 3.1. Post-Traumatic Growth Levels and Assessment Time on Coping Strategies

The influence of initial post-traumatic growth level on the use of coping strategies was studied. The total sample was divided into three groups according to their PGI-21 total score at T1: 85 patients with low post-traumatic growth (32.7%; 0–36 points), 84 with medium post-traumatic growth (32.3%; 37–60 points), and 91 with high post-traumatic growth (35%; 61–97 points).

One-way ANOVA and Chi square test analyses did not find any significant difference in sociodemographic and clinical sample characteristics between the three groups’ post-traumatic growth levels (Table 1).

#### 3.1.1. Effects of Post-Traumatic Growth and Assessment Time on Coping Strategies

As is presented in Table 2 and Figure 2, interaction effects between the initial level of post-traumatic growth and the assessment time were found for the following coping strategies: emotional support [F (4,514) = 4.252, *p* < 0.0001], positive reframing [F (4,514) = 4.395, *p* < 0.0001], active coping [F (4,514) = 6.612, *p* < 0.0001], and planning [F (4,514) = 5.107, *p* < 0.0001]. Simple effects show that these four coping strategies were used less frequently at T1 than T2 and T3 (d ranging from 0.58 to 1.03, medium to large effect size), and were more frequently used by patients in the high PTG group than in the low PTG group (d ranging from 0.29 to 0.43, small effect size). See Table 3.

Comparisons between different follow-ups in the three PTG groups are reported in Table 4. The variable time showed significant effects on these coping strategies in the low PTG group (d ranging from 0.45 to 1.24, small to large effect size), except for positive reframing from T2 to T3 (*p* = 0.529, d = 0.15). In addition, time also had an effect in the medium PTG group (ds from 0.61 to 1.06, medium to large effect size) and the high PTG group (d ranging from 0.42 to 0.67) when comparing the use of the four strategies at T1 with T2 and T3. Apart from planning (*p* = 0.048, d = 0.29), no significant difference was found between T2 and T3 in the use of these strategies in the medium and high PTG groups.

#### 3.1.2. Main Effects: Post-Traumatic Growth Level Effect on Coping Strategies

Regarding main effects, initial post-traumatic growth level was significant for instrumental support [F (2,257) = 5.107, *p* = 0.007] and self-distraction [F (2,257) = 3.319, *p* = 0.038] (Table 2).

Simple effect results reported in Table 3 indicate that instrumental support and self-distraction were significantly more frequent in the high PTG group compared to the low PTG group (d = 0.40 and 0.35, small effect size).

#### 3.1.3. Main Effects: Assessment Time on Coping Strategies

The time factor was also significant for acceptance [F (2,514) = 26.463, *p* < 0.0001], humor [F (2,514) = 82.970, *p* < 0.0001], religion [F (2,514) = 9.566, *p* < 0.0001], instrumental support [F (2,514) = 15.780, *p* < 0.0001], self-distraction [F (2,514) = 49.652, *p* < 0.0001], self-blame [F (2,514) = 109.678, *p* < 0.0001], and venting [F (2,514) = 24.560, *p* < 0.0001] (Table 2). These coping strategies were used less frequently at T1 compared to T2 and T3 (d ranging from 0.20 to 1.02, from small to large effect size). For self-distraction (*p* = 0.03, d = 0.26) and self-blaming (*p* < 0.0001, d = 0.28), the comparison between T2 and T3 was also significant, with a small effect size (Table 3).

### 3.2. Predictors of Post-Traumatic Growth

The total post-traumatic growth score was regressed on coping strategies at the three time points.

As is presented in Table 5, a higher score in positive reframing (β = 0.188, *p* < 0.001), emotional support (β = 0.152, *p* = 0.002), planning (β = 0.134, *p* = 0.008), religion means (β = 0.115, *p* = 0.017), and self-distraction (β = 0.098, *p* = 0.044) predicted greater PGI-21 total score levels at T1. These five coping strategies explained 20.1% of PGI-21 total score variance with a medium effect size (*f^2^* = 0.251).

In descending order of contribution, religion means (β = 0.281, *p* < 0.001), emotional support (β = 0.255, *p* < 0.001), active coping (β = 0.164, *p* = 0.001), positive reframing (β = 0.119, *p* = 0.023), and self-distraction (β = 0.107, *p* = 0.037) positively predicted PGI-total score at T2. All variables included in the model accounted for 29.4% of PGI-21 total score variance. The effect size coefficient indicated a large effect size (*f^2^* = 0.416) (Table 6).

At T3, the coping strategies that predicted the PGI-21 total score were positive reframing (β = 0.269, *p* < 0.001), instrumental support (β = 0.215, *p* < 0.001) and religion (β = 0.158, *p* = 0.006). A higher use of positive reframing, instrumental support, and religion predicted higher PGI-21 levels. The three variables together accounted for 17.2% of PGI-21 total score variance at T3, with a medium effect size (*f^2^* = 0.207) (Table 7).

## 4. Discussion

### 4.1. Post-Traumatic Growth Levels and Assessment Time on Coping Strategies

#### 4.1.1. Interaction Effects

In our study, we could identify a specific interaction between post-traumatic growth and different time points, showing that four different coping strategies were more often used over the course of time in patients with higher post-traumatic growth. Emotional support, positive reframing, active coping, and planning are these specific strategies, which help to adapt to the challenging situation and promote trauma processing and inner development.

Our findings broadly support previous research in the area connecting PTG with coping strategies. Regarding emotional support, it is proven that having a safe space to share emotions and communicate personal experiences with others facilitates the processing of a traumatic event [7]. Thus, participants of a therapeutic program promoting communication and emotional support between MS family members experienced a PTG increase [29]. The beneficial effect of seeking connection and contact with other family members in MS adaptation is supported by prior research [30].

Positive reframing is also a well-proven facilitator of post-traumatic growth [7,8,9]. It is a meaning-oriented coping strategy that modifies the way a situation is viewed, thereby positively changing its significance. It is a key element in Cognitive Behavioral Therapy (CBT). In agreement with the present results, previous studies related positive reframing with positive change and personal gain [16,17,18] in MS. Similar results were found in acquired brain damage [15].

Active coping and planning related to PTG have previously been found in other health conditions. As mentioned in the introduction, coping focused on actively resolving problems was associated with higher PTG in liver transplantation [14] and cervical dystonia [11]. Active coping, problem resolution, and planning are consistently related to positive outcomes in MS patients [3]. Planning and the proactive handling of challenging situations can support an inner conviction of being in control, thereby fueling self-efficacy. In particular, when facing situations where symptoms and disease progression are unpredictable and there is little to do on patients’ behalf, this active approach can be encouraging.

The increase in the use of these coping strategies over the 36-month follow-up period can indicate a gradual acquisition of a more adaptive coping style over the course of disease. This improvement can be seen as an encouraging sign regarding the possibility of a modification of coping using active training, especially because patients in the low PTG group showed a great increase in the utilization of adaptive coping styles from T1 to T3 with large effect sizes.

#### 4.1.2. Main Effect of Post-Traumatic Growth Level on Coping Strategies

Regardless of the assessment time, people with a higher PTG level used more instrumental support and self-distraction. The relevance of instrumental support was also found in patients with acquired brain damage [15], liver transplant recipients [14], and people with disabilities [10]. It is worthwhile to recognize the benefit of asking actively for help. It implies patients’ recognition of their own needs and a possible expression of the emotional discomfort entailed, which favors the development of PTG. Moreover, the assistance of others in managing daily activities can make patients’ everyday life easier [7,9].

The higher use of self-distraction in patients with higher PTG is rather surprising, as this strategy belongs to avoidance coping, which has been related to negative outcomes in MS according to an extensive systematic review in the field [3]. However, in other diseases, it has been proven that problem-focused as well as avoidance coping strategies may contribute to PTG. Thus, in acquired brain damage, greater use of self-distraction and problem-focused coping was associated with higher PTG [15]. In people with disabilities, problem-focused strategies and avoidance strategies were positive predictors of PTG. A longitudinal study comparing the impact of approach and avoidance coping found that in people recently diagnosed with a spinal cord injury (SCI), both coping styles contributed to PTG [31]. It is reasonable to argue that in the use of coping strategies the flexibility to react to specific situational demands is relevant for their success. The ability to distract oneself from frustration in uncontrollable situations might help patients to remain mentally stable and spare resources.

#### 4.1.3. Effect of Assessment Time on Coping Strategies

The factor assessment time determined the use of acceptance, humor, religion, instrumental support, self-distraction, self-blame, and venting. All these strategies were used less frequently in the first assessment.

The more frequent use of acceptance strategies in later disease stages is in line with findings in previous studies. Normally, the first response to a severe disease is denial, which provides time to become aware of the new situation and gently adopt the ability to confront oneself with the new reality and accept it [16].

Humor is another coping strategy that enables the patient to distance himself from an overwhelming situation. The development of this strategy may require some time, as humor is the result of a long process of cognitive reconstruction and rumination about a traumatic event [32,33]. The greater use of religion in the later follow-ups can be explained by the age-associated increase in spirituality and religiosity [34].

The more frequent use of instrumental support can go along with a greater use of acceptance and humor. It is therefore likely that when patients present a higher acceptance of their condition and can even make jokes about it, they feel more confident to ask for help. Accepting the reality of a difficult situation implies the recognition that it must be addressed [8], and asking for a helping hand is a manner of tackling it.

The observed increase in the use of self-distraction could be connected with disease progression. The loss of functioning over time and an increasing number of uncontrollable, unpleasant situations may require a greater demand to focus the mind on other activities. In keeping with this assumption, preceding studies indicate a relation between avoidant coping and a more severe MS clinical profile [35,36].

Regarding the use of venting, the increase over time could be explained by the need for practice. The conceptualization of venting in the COPE-28 questionnaire implies active efforts to release unpleasant emotions [37]. Therefore, venting requires time, as patients first have to identify negative emotions and then find a way to let them out.

Self-blame also showed an increase over the course of the study, which is in keeping with previous findings in MS [38].It is important to consider that self-blame was also pointed out as a risk factor for quality of life in MS. A higher frequency of self-blame has been observed in liver transplant patients compared to their caregivers [14]. One might argue that feelings of shame and guilt may arise from disease progression, impaired functioning, and disability. A previous study found that disease-related shame levels in adults with MS are connected to withdrawal and self-blame [38].

### 4.2. Predictors of Post-Traumatic Growth

This study also investigated the prediction of PTG by coping strategies. Consistent with the literature, positive reframing and religion were unveiled as positive predictors of PTG across all three assessment times. A meta-analysis concluded that religious coping and positive reappraisal were the most consistent predictors of PTG [8].The possibilities of growth from suffering and crisis lies at the heart of all religious beliefs [39]. Religious beliefs encourage positive reframing by reappraising hopeless situations as challenging tasks [9]. In addition, religion fosters social support through social relationships in the religious community or participation in social activities [7].

Positive reframing was identified as a predictor of PTG in other chronic diseases, such as acquired brain injury [15] and hemodialysis [12]. The importance of a positive reappraisal of the current situation in MS has not only been reported in patients [16,17,18], but also in families and caregivers [29,30,40].

Seeking social support was also a protective factor of PTG across all time points. It appeared as emotional support in the first two assessments and in the last follow-up as instrumental support. As patients had been diagnosed with MS an average of twelve years ago, the vast majority had ample experience with the different facets of MS. Thus, one might argue that this shift may mirror growing functional impairment with disease progression, which is associated with greater emphasis on the need for practical help, alongside a growing awareness of one’s needs and an ability to communicate them. This ability implies a greater sense of intimacy and freedom to be oneself, disclosing even socially undesirable aspects to others [39]. These results are consistent with growing evidence that having a safe social context to express feelings facilitates the disease experience [9]. Actually, seeking and accepting social support is a therapeutic target when working with MS patients and their families in stress coping [24].

Self-distraction positively predicted post-traumatic growth in the first two assessments. This result is in line with a longitudinal study of spinal cord injury patients proposing that turning one’s attention from an unpleasant mental or physical state to comforting or neutral activities is a significant facilitator of personal growth [26]. Self-distraction has also been shown as a protective factor for PTG in people with physical disabilities [10]. The ability of MS patients to distract themselves from distress may be a significant coping strategy to enable personal growth in specific phases [7].

Based on our findings, the following therapeutic interventions should be considered in clinical practice to enhance PTG in MS. CBT aims to identify and modify distorted thoughts using positive reframing and can support this specific coping style. Active coping and planning to overcome obstacles and increase self-efficacy are further techniques supported by CBT [3,37].

Acceptance and Commitment Therapy (ACT) is a modern form of CBT and aims to accept unpleasant feelings such as self-blame rather than eliminating them. Patients might learn to have a more open approach to uncomfortable feelings and avoid maladaptive coping [19].

A wide range of therapies are available and proven to improve emotional and instrumental support, such as social skills training and self-help groups. Social skills training strengthens social relationships by promoting communication and recognition of significant others’ needs. Self-help groups facilitate the empathic sharing of emotions in a socially safe environment [3,37].

### 4.3. Limitations

The current study shows some weaknesses and strengths. The non-random selection of participants limits the external validity of the study. In addition, there is a risk of social desirability bias due to the use of self-report questionnaires. Nevertheless, the large sample size, the longitudinal design with a 36-month follow-up period, and the low dropout rate are major strengths of this study.

## 5. Conclusions

The present study provides data relating PTG and coping strategies in MS. The positive impact of specific coping strategies such as positive reframing, social support, active coping, planning, and religion on PTG speaks for the relevance of supporting these strategies from the beginning of the disease. Furthermore, the relevance of self-distraction as a protective PTG factor in MS points to the usefulness of flexibility when coping with situational demands. Therapeutic interventions should train the awareness of internal and external needs to enable patients to deal with them in a more adaptive manner. In addition, there was an increase in adaptive coping over time, which suggests an improved adjustment to the disease, thereby facilitating PTG. On the other hand, self-blame also increased over time, which poses a risk to PTG. Thus, special attention should be taken to recognize and address the usage of maladaptive coping early on.

## Figures and Tables

**Figure 1 ijerph-19-12679-f001:**
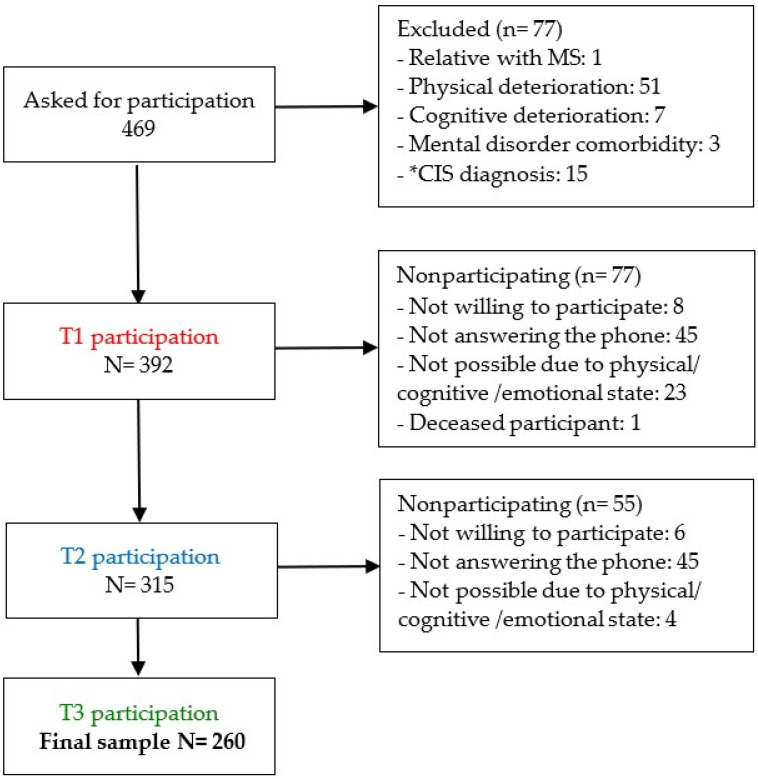
Study flow-chart. * Note. CIS = Clinically isolated syndrome. The sample selection process at the three different periods in different colors.

**Figure 2 ijerph-19-12679-f002:**
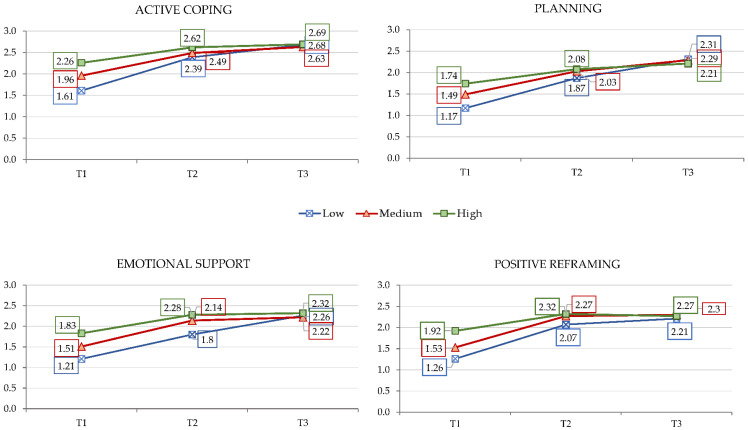
Interaction effects between initial level of post-traumatic growth and evaluation phases.

**Table 1 ijerph-19-12679-t001:** Comparison of sociodemographic and clinical characteristic of the three different post-traumatic growth level groups at T1.

	Post-Traumatic Growth Level			Intergroup Comparisons		Effect Size
	Low (n = 85)	Medium (n = 84)	High (n = 91)	χ^2^	*p*	Cohen´s *w*
Gender n (%)				0.222	0.895	0.029 (N)
Male	28 (32.9)	26 (31)	27 (29.7)			
Female	57 (67.1)	58 (69)	64 (70.3)			
Partnership n (%)				1.871	0.392	0.084 (N)
No partner	59 (69.4)	66 (78.6)	68 (74.7)			
Partner	26 (30.6)	18 (21.4)	23 (25.3)			
Occupation n (%)				1.247	0.536	0.068 (N)
Employed/In education	34 (40)	27 (32.1)	31 (34.1)			
Unemployed	51 (60)	57 (67.9)	60 (65.9)			
Educational level n (%)				8.039	0.090	0.175 (S)
Primary education	8 (9.4)	10 (11.9)	17 (18.7)			
Secondary education	26 (30.6)	34 (40.5)	22 (24.2)			
University or higher	51 (60)	40 (47.6)	52 (57.1)			
MS subtype n (%)				0.541	0.763	0.046 (N)
Remittent	76 (89.4)	72 (85.7)	80 (87.9)			
Progressive	9 (10.6)	12 (14.3)	11 (12.1)			
				**F (2,257)**	** *p* **	**Cohen´s *d***
Age (M ± SD)	45.7 ± 11.1	45.4 ± 10.9	44.1 ± 9.9	0.581	0.560	0.015 (N)
EDSS (M ± SD)	3.1 ± 1.8	3.1 ± 1.9	3.4 ± 2.1	0.704	0.495	0.040 (N)
Months since diagnosis (M ± SD)	140.3 ± 98.4	147.6 ± 92.3	146.2 ± 77.8	0.159	0.853	0.021 (N)
Months since outbreak (M ± SD)	181.4 ± 112.9	190.8 ± 112.4	184.4 ± 108.4	0.084	0.919	0.020 (N)

Table note: S = small effect size, N = effect size.

**Table 2 ijerph-19-12679-t002:** Coping strategies: differences in coping strategies use by evaluation phase and initial post-traumatic growth. (3 × 3 mixed factorial analysis of variance).

	Main Effects (Cohen´s *f*)		
	Phase	Post-Traumatic Growth	Interaction Effects (Cohen´s *f*)
	F (2,514)	F (2,257)	F (4,514)
Acceptance	26.463 ** 0.320 M	2.869 0.149 S	0.378 0.054 N
Emotional support	82.898 ** 0.568 L	7.050 ** 0.234 S	4.252 ** 0.181 S
Humor	82.970 ** 0.568 L	4.623 0.190 S	1.383 0.105 S
Positive reframing	75.263 ** 0.541 L	5381 ** 0.204 S	4.395 ** 0.184 S
Religion	9.566 ** 0.193 S	1.019 0.089 N	1.239 0.1 S
Active coping	102.369 ** 0.631 L	11.2 ** 0.29 M	6.612 ** 0.226 S
Instrumental support	15.780 ** 0.250 M	5.107 ** 0.198 S	0.923 0.083 N
Planning	86.358 ** 0.580 L	3.239 * 0.206 S	5.107 ** 0.198 S
Behavioral disengagement	0.015 0 N	1.091 0.008 N	1.393 0.011 N
Denial	0.325 0.031 N	1.298 0.100 S	3.035 0.153 S
Self-distraction	49.652 ** 0.439 L	3.319 * 0.160 S	0.780 0.077 N
Self-blaming	109.678 ** 0.653 L	0.767 0.077 N	0.364 0.054 N
Substance use	1.008 0.063 N	0.792 0.044 N	1.012 0.089 N
Venting	24.560 ** 0.308 M	0.736 0.077 N	1.246 0.101 S

Table note: L = large effect size, M = medium effect size, S = small effect size, N = null effect size. Significance value * *p* < 0.05, ** *p* < 0.001.

**Table 3 ijerph-19-12679-t003:** Coping strategies: differences between coping strategies used according to post-traumatic growth level and evaluation phases.

	Post-Traumatic Growth Level M (SD)			Phases			Comparisons *p* (Cohen’s *d*)					
	Low (a) n = 85	Medium (b) n = 84	High (c) n = 91	1	2	3	Group Levels			Phases		
							a–b	a–c	b–c	1–2	1–3	2–3
Acceptance	2.40 (0.81)	2.44 (0.72)	2.56 (0.57)	2.28 (0.73)	2.58 (0.56)	2.56 (0.60)	1 0.05 N	0.072 0.23 S	0.237 0.18 N	<0.001 0.58 M	<0.001 0.42 S	1 0.03 N
Emotional support	1.75 (0.95)	1.96 (0.77)	2.15 (0.91)	1.52 (0.92)	2.07 (0.96)	2.26 (0.83)	0.178 0.24 S	0.001 0.43 S	0.211 0.23 S	<0.0001 0.58 M	<0.0001 0.88 L	0.005 0.21 S
Humor	1.81 (1.05)	1.83 (1.04)	2.13 (1.10)	1.42 (1.10)	2.24 (0.98)	2.11 (1.02)	1 0.02 N	0.023 0.30 S	0.035 0.28 S	<0.0001 0.79 M	<0.0001 0.65 M	0.103 0.13 N
Positive reframing	1.84 (0.95)	2.03 (0.81)	2.17 (0.93)	1.57 (0.94)	2.21 (0.88)	2.26 (0.87)	0.196 0.22 S	0.004 0.30 S	0.504 0.16 N	<0.0001 0.70 M	<0.0001 0.76 M	1 0.06 N
Religion means	0.90 (0.88)	1.06 (0.73)	1.07 (0.94)	0.86 (0.98)	1.06 (1.07)	1.09 (1.08)	0.701 0.19 N	0.605 0.19 N	1 0.01 N	0.001 0.20 S	<0.0001 0.22 S	1 0.03 N
Active coping	2.25 (0.85)	2.36 (0.65)	2.52 (0.67)	1.94 (0.79)	2.50 (0.62)	2.67 (0.58)	0.121 0.15 N	<0.0001 0.35 S	0.029 0.24 S	<0.0001 0.78 M	<0.0001 1.03 L	0.004 0.28 S
Instrumental support	1.42 (0.78)	1.55 (0.84)	1.76 (0.90)	1.38 (0.86)	1.64 (0.98)	1.73 (0.96)	0.713 0.16 N	0.005 0.40 S	0.158 0.24 S	<0.0001 0.28 S	<0.0001 0.38 S	0.555 0.09 N
Planning	1.78 (0.78)	1.94 (0.76)	2.01 (0.82)	1.47 (0.83)	1.99 (0.90)	2.27 (0.81)	0.293 0.20 S	0.039 0.29 S	1 0.09 N	<0.0001 0.60 M	<0.0001 0.98 L	<0.001 0.33 S
Behavioral disengagement	0.39 (0.57)	0.37 (0.52)	0.40 (0.66)	0.39 (0.59)	0.39 (0.61)	0.38 (0.54)	1 0.04 N	1 0.02 N	0.427 0.05 N	1 0 N	1 0.02 N	1 0.02 N
Denial	0.46 (0.62)	0.45 (0.73)	0.36 (0.57)	0.41 (0.64)	0.42 (0.61)	0.45 (0.63)	1 0.02 N	0.469 0.17 N	0.535 0.14 N	1 0.02 N	1 0.06 N	1 0.05 N
Self-distraction	1.40 (1.05)	1.51 (0.92)	1.77 (1.07)	1.57 (1.02)	2.05 (0.94)	2.28 (0.78)	0.552 0.11 N	0.032 0.35 S	0.678 0.26 S	<0.0001 0.50 M	<0.0001 0.78 M	0.003 0.26 S
Self-blaming	1.61 (0.99)	1.74 (0.88)	1.71 (1.02)	1.15 (0.97)	1.83 (0.90)	2.07 (0.81)	0.703 0.14 N	1 0.09 N	1 0.03 N	<0.0001 0.77 M	<0.0001 1.02 L	<0.0001 0.28 S
Substance use	0.10 (0.35)	0.113 (0.23)	0.13 (0.47)	0.11 (0.37)	0.14 (0.51)	0.10 (0.35)	1 0.04 N	0.742 0.07 N	1 0.05 N	<0.0001 0.07 N	<0.0001 0.02 N	0.705 0.09 N
Venting	0.97 (0.83)	0.89 (0.68)	1.15 (0.81)	1.01 (0.78)	1.26 (0.80)	1.37 (0.64)	1 0.11 N	1 0.12 N	0.766 0.35 S	<0.0001 0.32 S	<0.0001 0.50 M	0.138 0.15 N

Table note: L = large effect size, M = medium effect size, S = small effect size, N = null effect size.

**Table 4 ijerph-19-12679-t004:** Coping strategies: differences in coping strategies use between evaluation phases by initial post-traumatic growth level.

	Comparison between Phases, *p* (Cohen´s *d*)								
	Low CP			Medium CP			High CP		
	1–2	1–3	2–3	1–2	1–3	2–3	1–2	1–3	2–3
Emotional support	<0.0001 0.61 M	<0.0001 1.06 L	<0.0001 0.50 M	<0.0001 0.72 M	<0.0001 0.86 L	1 0.09 N	<0.0001 0.49 S	<0.0001 0.57 M	1 0.05 N
Positive reframing	<0.0001 0.85 L	<0.0001 1.05 L	0.529 0.15 N	<0.0001 0.87 L	<0.0001 0.90 L	1 0.02 N	0.002 0.46 S	0.003 0.39 S	1 0.05 N
Active coping	<0.0001 0.85 L	<0.0001 1.24 L	<0.0001 0.45 S	<0.0001 0.78 M	<0.0001 1.06 L	0.318 0.23 S	<0.0001 0.60 M	<0.0001 0.65 M	1 0.08 N
Planning	<0.0001 0.79 M	<0.0001 1.18 L	<0.0001 0.50 M	<0.0001 0.61 M	<0.0001 1.01 L	0.048 0.29 S	0.003 0.42 S	<0.0001 0.67 M	0.642 0.22 S

Table note: L = large effect size, M = medium effect size, S = small effect size, N = null effect size.

**Table 5 ijerph-19-12679-t005:** T1 Post-traumatic growth (CP-21) Multiple linear regression model.

	F	R^2^	B	SE.B	β	1-β	*f^2^*
Model 1	54.523 (1.405)	0.119 **	34.292	2.159		0.99	0.135 (S)
Positive reframing			8.798	1.192	0.344 **		
Model 2	37.997 (2.404)	0.158 **	28.824	2.456		0.99	0.187 (M)
Positive reframing			7.001	1.236	0.274 **		
Emotional support			5.402	1.238	0.211 **		
Model 3	29.391 (3.403)	0.180 **	25.331	2.659		0.99	0.222 (M)
Positive reframing			5.869	1.272	0.230 **		
Emotional support			4.604	1.248	0.180 **		
Planning			4.471	1.386	0.158 **		
Model 4	24.082 (4.402)	0.193 *	24.812	2.647		0.99	0.239 (M)
Positive reframing			5.306	1.281	0.208 **		
Emotional support			3.791	1.277	0.148 **		
Planning			4.457	1.376	0.158 **		
Religion means			3.054	1.165	0.126 **		
Model 5	20.227 (5.401)	0.201 *	22.896	2.802		0.99	0.251 (M)
Positive reframing			4.806	1.300	0.188 **		
Emotional support			3.883	1.273	0.152 **		
Planning			3.784	1.410	0.134 **		
Religion means			2.789	1.168	0.115 *		
Self-distraction			2.361	1.170	0.098 *		

Table note: M = medium effect size, S = small effect size. Significance value * *p* < 0.05, ** *p* < 0.001.

**Table 6 ijerph-19-12679-t006:** T2 Post-traumatic growth (CP-21) Multiple linear regression model.

	F	R^2^	B	SE.B	β	1-β	*f^2^*
Model 1	43.189 (1.322)	0.118 **	52.846	1.614		0.99	0.134 (S)
Religion means			7.356	1.119	0.344 **		
Model 2	47.097 (2.323)	0.227 **	36.552	2.860		1	0.294 (M)
Religion means			7.094	1.050	0.332 **		
Emotional support			7.876	1.173	0.330 **		
Model 3	38.739 (3.321)	0.266 **	21.330	4.606		1	0.362 (L)
Religion means			6.955	1.025	0.325 **		
Emotional support			6.985	1.164	0.292 **		
Active coping			6.927	1.668	0.202 **		
Model 4	31.681 (4.320)	0.284 **	18.040	4.703		1	0.396 (L)
Religion means			6.388	1.034	0.299 **		
Emotional support			6.414	1.169	0.269 **		
Active coping			5.793	1.698	0.169 **		
Positive reframing			3.586	1.270	0.144 **		
Model 5	26.496 (5.319)	0.294 *	15.609	4.820		1	0.416 (L)
Religion means			6.011	1.044	0.281 **		
Emotional support			6.084	1.174	0.255 **		
Active coping			5.616	1.691	0.164 **		
Positive reframing			2.956	1.298	0.119 *		
Self-distraction			2.608	1.242	0.107 *		

Table note: L = large effect size, M = medium effect size, S = small effect size. Significance value * *p* < 0.05, ** *p* < 0.001.

**Table 7 ijerph-19-12679-t007:** T3 Post-traumatic growth (CP-21) Multiple linear regression model.

	F	R^2^	B	SE.B	β	1-β	*f^2^*
**Model 1**	104.556 (1.312)	0.093 **	60.810	3.252		0.98	0.103 (S)
Positive reframing			7.037	1.344	0.306		
**Model 2**	56.012 (2.311)	0.148 **	53.024	3.679		0.99	0.173 (M)
Positive reframing			6.801	1.306	0.295		
Instrumental support			4.831	1.172	0.234		
**Model 3**	39.010 (3.310)	0.172 **	51.892	3.657		1	0.207 (M)
Positive reframing			6.198	1.309	0.269		
Instrumental support			4.434	1.166	0.215		
Religion			2.866	1.038	0.158		

Table note: M = medium effect size, S = small effect size. Significance value ** *p* < 0.001.

## Data Availability

The raw data upholding the conclusions of this research will be made available by the authors upon request.

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
