# Peer review of "Coping Strategies Furthering Post-Traumatic Growth in Multiple Sclerosis: A Longitudinal Study"

_ijerph, 2022, doi:10.3390/ijerph191912679_

Round 1
Reviewer 1 Report
The authors present a prospective study to evaluate post traumatic growth (PTG) in MS patients. The results of the study are nicely presented. I have a few comments:
1. In the introduction, the authors mention that there is a lack of studies investigating the longitudinal aspects of PTG in MS. While this may be correct, the presently available studies should be discussed in detail and referenced.
2. Please describe briefly the coping strategy 'active coping'.
3. Line 309-311: The authors state that self-distraction has been associated with negative outcomes in MS in many studies. However, only one reference is cited.
4. In the discussion, the authors state that some coping strategies (self-distraction, seeking emotional support) may be related to disease progression. Has the disease progression been captured in this longitudinal time frame and could be linked with the findings to further substantiate these assumptions?
5. Considering this study's longitudinal design in the context of the lengthy disease process of MS, 36 months is certainly a good follow-up time. However, it may be interesting to watch for follow-up data in the future. Do the authors plan on gathering follow-up data?
6. From a therapeutic perspective, what are the most effective therapeutic strategies to promote the coping strategies identified to further PTG in MS patients. Please discuss therapeutic options and enlighten the findings from a clinical/therapeutic perspective.
7. Instead of ending the conclusion section with the study's limitations, please move the limitations to the discussion as a separate subsection.
Author Response
The authors present a prospective study to evaluate post traumatic growth (PTG) in MS patients. The results of the study are nicely presented. I have a few comments:
ANSWER: Thank you for the encouraging answer.
- In the introduction, the authors mention that there is a lack of studies investigating the longitudinal aspects of PTG in MS. While this may be correct, the presently available studies should be discussed in detail and referenced.
ANSWER: Thanks for the useful comment. We revised the introduction accordingly and discussed and referenced the available papers studying longitudinal aspect of PTG.
- Please describe briefly the coping strategy 'active coping'.
ANSWER: 'Active coping' has been briefly described the first time it appears in the introduction. If there is any other location where it suits better, please just let us know.
- Line 309-311: The authors state that self-distraction has been associated with negative outcomes in MS in many studies. However, only one reference is cited.
ANSWER: Thank you for this hint. The studies finding this association are included in the referenced systematic review. We revised the section accordingly.
- In the discussion, the authors state that some coping strategies (self-distraction, seeking emotional support) may be related to disease progression. Has the disease progression been captured in this longitudinal time frame and could be linked with the findings to further substantiate these assumptions?
ANSWER: According to clinical records we made the general observation that EDSS tends to increase with time. However, due to different assessment times for patients’ self-report questionnaires and the EDSS disability evaluation our findings cannot further substantiate our assumptions.
- Considering this study's longitudinal design in the context of the lengthy disease process of MS, 36 months is certainly a good follow-up time. However, it may be interesting to watch for follow-up data in the future. Do the authors plan on gathering follow-up data?
ANSWER: It would have been extremely interesting and productive to include further follow-ups. However, due to different limitations inherent in the research project an extension of the time frame is not possible.
- From a therapeutic perspective, what are the most effective therapeutic strategies to promote the coping strategies identified to further PTG in MS patients. Please discuss therapeutic options and enlighten the findings from a clinical/therapeutic perspective.
ANSWER: Thank you for this helpful comment. As suggested different therapeutic options have been discussed in the discussion section.
- Instead of ending the conclusion section with the study's limitations, please move the limitations to the discussion as a separate subsection.
ANSWER: Limitations have been moved to a separate subsection in the discussion section.
Even though it is not marked in the document; English language had been revised.
Please see the attachment to view the answer to your revision in a PDF file.

Reviewer 2 Report
I found the article very interesting and informative. Variables were identified from the beginning of the introduction. The gap in the literature was clearly presented, leading to the purpose of the study.
Line 64: What are the criteria for MS inclusion, and how is the diagnosis verified? Medical records, word of mouth from participants???
Line 126: EDSS, would you consider it informative to include a little information for the reading audience as to what EDSS stands for, how test results are reached, and what those results mean to participants and medical providers? (Expanded Disability Status Scale (EDSS) is the most used scale in multiple sclerosis (MS) patients. EDSS is a very effective method of reflecting disability).
The reference section included articles offering current information (< five years) for this study. Kudos.
References:
Haber A, LaRocca NG. eds. Minimal Record of Disability for multiple sclerosis. New York: National Multiple Sclerosis Society; 1985.
Kurtzke J. F. Rating neurologic impairment in multiple sclerosis: an expanded disability status scale (EDSS). Neurology. 1983 Nov;33(11):1444-52.
Şen, S. Neurostatus and EDSS Calculation with Cases. Arch Neuropsychiatry 2018;55: (Supplement 1): S80−S83 https://doi.org/10.29399/npa.23412
Author Response
I found the article very interesting and informative. Variables were identified from the beginning of the introduction. The gap in the literature was clearly presented, leading to the purpose of the study.
ANSWER: Thanks for the encouraging comment.
Line 64: What are the criteria for MS inclusion, and how is the diagnosis verified? Medical records, word of mouth from participants???
ANSWER: All included patients were diagnosed according to McDonald criteria. The primary neurologist confirmed the diagnosis. This procedure has been included in the participants and procedure section 2.1..
Line 126: EDSS, would you consider it informative to include a little information for the reading audience as to what EDSS stands for, how test results are reached, and what those results mean to participants and medical providers? (Expanded Disability Status Scale (EDSS) is the most used scale in multiple sclerosis (MS) patients. EDSS is a very effective method of reflecting disability).
ANSWER: Thank you for the useful comment and references. In section 2.2.3. the Expanded Disability Status Scale (EDSS) has been inserted and relevant information about the EDSS scale are presented.
Even though it is not marked in the document; English language had been revised.
